# Preparation and Characterization of Hydrophobically Grafted Starches by In Situ Solid Phase Polymerization

**DOI:** 10.3390/polym11010072

**Published:** 2019-01-05

**Authors:** Yingfeng Zuo, Xiaoyu He, Ping Li, Wenhao Li, Yiqiang Wu

**Affiliations:** College of Materials Science and Engineering, Central South University of Forestry and Technology, Changsha 410004, Hunan, China; zuoyf1986@163.com (Y.Z.); h2221810261@163.com (X.H.); pzxiaomei@163.com (P.L.); leewh1994@163.com (W.L.)

**Keywords:** corn starch, maleic anhydride, lactic acid, methyl acrylate, hydrophobic modification, in situ solid phase polymerization

## Abstract

Three kinds of hydrophobic groups grafted starches of maleic anhydride grafted starch (MAH-*g*-starch), lactic acid grafted starch (LA-*g*-starch), and methyl acrylate grafted starch (MA-*g*-starch) were prepared by in situ solid phase polymerization. The results of Fourier transform infrared spectroscopy (FT-IR) and X-ray photoelectron spectroscopy (XPS) confirmed successful grafting. The grafting ratios of MAH-*g*-starch, LA-*g*-starch, and MA-*g*-starch were 6.50%, 12.45%, and 0.57%, respectively. Influenced by the grafting ratio, LA-*g*-starch had the best relative hydrophobicity and the largest molecular weight, and those for MA-*g*-starch were the worst. The surfaces of grafted starches were covered with graft polymer, with obvious surface roughness and bond degree of MAH-*g*-starch and LA-*g*-starch. The crystalline structure of grafted starches showed some damage, with LA-*g*-starch exhibiting the greatest decrease in crystallinity, and less of a change for MA-*g*-starch. Overall, the grafting reaction improved thermoplasticity, with LA-*g*-starch the most improved, followed by MAH-*g*-starch, and then MA-*g*-starch.

## 1. Introduction

In recent years, with increasing environmental awareness and the gradual consumption of fossil resources, there has been significant interest in the development of green and renewable biomass resources as a substitute for petrochemical products and as alternate energy sources. Starch is a natural polymer material with many advantages such as a wide variety of sources, low price, biodegradability, and renewability [1,2,3]. Thus, starch has received increased interest and is used in the production of thermoplastic starch plastics and starch/polymer blend composites [4,5]. However, starch is a hydrophilic polymer due to its large number of hydrophilic hydroxyl groups, making thermoplastic starch plastics susceptible to moisture attacks and significant changes in dimensional stability and mechanical properties [6]. This hydrophilicity also leads to poor interfacial compatibility between starch and the hydrophobic polymer, causing poor properties in the starch/polymer blend composites [7]. Reaction of the hydrophilic hydroxyl groups on starch to decrease their number or replacement of the hydroxyl groups with hydrophobic groups can significantly improve the relative hydrophobicity of starch, allowing a more compatible interface with the polymer.

One strategy for modification is the use of hydrophobic groups to replace hydrophilic hydroxy groups on the starch molecule chain [8,9]. To do this, most strategies have focused on the starch acetylating [10] and the grafting of highly reactive hydrophobic functional groups onto the starch polymeric backbone [11,12,13]. The grafted functional groups could react with the hydroxyl groups of starch macromolecules to form covalent bonds, providing better control of phase size and improved interfacial adhesion. Additionally, Sagar [14] believed that modification of starch could increase the length of the starch side chain, which enhanced thermoplastic and hydrophobic characteristics, and that the hydrophobic function group in the starch structure could play the role of plasticization.

Hydrophobic groups graft starches are usually produced by the wet method, organic solvent method [15], reactive extrusion method [16], or microwave-assisted method [17,18]. The wet method is performed in an aqueous solution, so the reaction is homogeneous and environmentally friendly. However, the hydrolytic side reaction of anhydride cannot be ignored in this process. The organic solvent method offers a homogeneous reaction, but has a low degree of substitution of products, a high output cost, and causes environmental pollution. The reactive extrusion method requires the addition of a plasticizer; this will make the starch undergo plasticization, changing the granular structure of the starch. The microwave-assisted method is not suitable for large-scale industrial production, as the process is relatively complex and requires significant energy consumption.

Considering the limitations of the current technological methods used to graft hydrophobic groups to starch, the objective of this investigation was to produce hydrophobic groups grafted to starches by an in situ solid phase polymerization method. In this process, dry starch, reaction monomers, and a catalyst are mixed in a closed hydrothermal reactor, and in situ polymerization of the starch and reaction monomers is initiated under pressure at 80 °C (Figure 1). This approach is a promising green production method with many advantages. First, this process uses a reaction temperature that is lower than traditional methods and produces fewer by-products and degradation products. Second, the reacting monomer is in complete contact with the starch, so the reaction is efficient. Third, the polycondensation reaction is stable and can be scaled to an industrial level. Fourth, the reaction is performed in the presence of solvents, for a more environmentally friendly condensation process. Furthermore, the pressure will be generated automatically in the reactor when the temperature reaches a certain range, to accelerate the reaction rate. Therefore, the proposed process of in situ solid phase polymerization of starch and reaction monomers is a new, facile, and effective way to improve the interfacial compatibility of hydrophilic starch and hydrophobic polymer.

Anhydrides, carboxylic acids, and acrylate compounds are commonly used in hydrophobic modification of starch and interfacial modification of starch/polymer matrix composites [19,20,21,22]. Among them, maleic anhydride (MAH), lactic acid (LA), methyl acrylate (MA), and acetic acid are most commonly used. This is mainly due to the fact that MAH, LA, and acetic acid have carboxyl groups and MA contains free radical polymerization of C=C, which can react with hydroxyl groups in starch. Therefore, to test this in situ solid phase polymerization method, maleic anhydride, lactic acid, and methyl acrylate were used as graft monomers to prepare hydrophobically modified starch. The grafting effect and hydrophobicity modification were then compared for the three kinds of grafting monomers so as to improve the application range of starch in the field of thermoplastic starch and starch/polymer composites.

## 2. Materials and Methods

### 2.1. Materials

Corn starch was obtained from Dacheng Corn Development Co. Ltd. (Changchun, Jilin, China), and dried in a vacuum drying oven of 50 °C for 48 h to eliminate moisture before use. Methyl acrylate was obtained from Tianjin Kwangfu Fine Chemical Industry Research Institute (Tianjin, China). Lactic acid and stannous octoate were supplied by Sinopharm Chemical Reagent Co., Ltd (Shanghai, China). Maleic anhydride, acetone, and ammonium persulfate were purchased from Tianjin Kemiou Chemical Reagent Co., Ltd. (Tianjin, China). All chemicals were AR grade.

### 2.2. Preparation of Hydrophobically Modified Starch

Thirty grams of corn starch were added into a hydrothermal reactor, together with 4.5 g grafting monomer and 0.9 g stannous octanoate catalyzer, and mixed and ground evenly. The hydrothermal reactor was sealed and placed in an oven at a set temperature (60, 70, 80, or 90 °C) for in situ solid phase polymerization. Heating was stopped after a set time (0.5, 1.0, 1.5, 2.0, or 2.5 h). When the reacted starches were cooled to room temperature, acetone was added to the mixture. The mixture was stirred and subjected to suction filtration. Finally, the mixture was washed three times with acetone and dried in an oven at 55 °C until a constant weight was achieved.

At the same time, in order to compare the grafting effect of in situ solid phase polymerization, the graft copolymerization of starch was carried out by aqueous phase polymerization and organic solvent polymerization. Deionized water was the solvent in aqueous phase polymerization, and tetrahydrofuran was selected by organic solvent polymerization. Starch and solvents were mixed into starch emulsion according to the mass ratio of 1/9, and protection of flushing nitrogen into reaction system. The reaction temperature was 80 °C, the reaction time was 2.0 h, and the catalyzer and treatment of grafted starches were consistent with in situ solid phase polymerization.

### 2.3. Properties and Characterization

#### 2.3.1. Fourier Transform Infrared Spectroscopy (FT-IR) Analysis

In order to characterize the chemical changes in the grafted starch, the samples were tabletted with KBr and subjected to FTIR (IRAffinity-1, Shimadzu, Kyoto, Japan). To completely remove the moisture, the native starch and hydrophobic group grafted starch materials were further dried in a muffle oven (at 50 °C) for 48 h. The tested samples were obtained after grinding fully using a weight ratio of sample: KBr of 1:100. The FTIR curves for the samples were obtained at a range of 400–4000 cm^−1^.

#### 2.3.2. X-ray Photoelectron Spectroscopy (XPS) Analysis

XPS measurements were performed at room temperature with monochromatic AlKα radiation (1486.6 eV) using a K-Alpha X-ray photoelectron spectrometer (supplied by Thermo Fisher Scientific Co., Ltd., Billerica, MA, USA). The X-ray beam was a 100 W, 200 mm-diameter beam raster over a 2 mm by 0.4 mm area on the sample. A high-energy photoemission spectrum was collected using a pass energy of 50 eV and resolution of 0.1 eV. For the Ag_3_d_5_/2 line, these conditions produced an FWHM of 0.80 eV. 

#### 2.3.3. Determination of Grafting Ratio 

First, 1.00 g of dry grafted starch was weighed and placed in a 250-mL conical flask. Next, 10 mL of 75% ethanol solution in deionized water were added to the flask, followed by the addition of 10 mL of 0.5 mol/L aqueous sodium hydroxide solution. The stoppered conical flask was agitated, warmed to 30 °C, and stirred for 1 h. The excess alkali was then back-titrated with a standard 0.5 mol/L aqueous hydrochloric acid solution. A blank titration was performed using native, un-modified starch. The grafting ratio (*GR*) was calculated as follows:$$W_{}=\frac{Mc(V_{0}-V_{1})}{1000\times nm}\times 100\%$$
$$GR=\frac{162W_{}}{M\times (100-W_{})}\times 100\%,$$
where *W* is the substitution degree of the hydrophobic group, %; *M* is the molecular weight of the graft monomer; *c* is the concentration of the aqueous hydrochloric acid solution, mol/L; *V*_0_ is the consumed volume of aqueous hydrochloric acid solution by the blank sample, mL; *V*_1_ is the consumed volume of aqueous hydrochloric acid solution by the grafted starch sample, mL; *n* is the number of hydrophobic groups from the grafted monomer; and *m* is the mass of the sample, g. 

#### 2.3.4. Contact Angle Measurement

The native starch and grafted starches were weighed, mixed, and pressed into a pie sample of 1.5 cm diameter with a press machine whose pressure was 20 MPa. An optical contact angle measurement instrument (Data Physics OCA20, DataPhysics Instruments, Filderstadt, Germany) was used to measure the contact angle of samples, using distilled water as the test solution. For each measurement, 4 µL liquid in a microsyringe were dripped on to the surface of the samples, and the contact angle values were measured until 0°.

#### 2.3.5. Determination of Water Absorption

To determine the water absorption of the grafted starches, 4.0 g native starch and grafted starches (dry base) were separately placed in glass dishes that contained a set amount of water. Over the test period the weight of each sample was measured every 12 h. The water absorption was calculated as follows:$$\mathrm{water}\;\mathrm{absorption}=\frac{W_{t}-W_{0}}{W_{0}}\times 100\%,$$
where *W_t_* was the weight of the sample after water absorption for *t* hours and *W*_0_ was the weight of the sample when it reached a dry constant weight. 

#### 2.3.6. Gel Permeation Chromatography (GPC) Measurement

The molecular weight of native starch, MAH-*g*-starch, LA-*g*-starch, and MA-*g*-starch were tested with a gel permeation chromatograph system (Viscotek TDA305max, Malvern Instruments Co., Ltd., Malvern, UK). The solvent was dimethyl sulfoxide (DMSO) + 20 mmol LiBr, the column set was IGuard + 1 × I-H, the flow rate was 0.500 mL/min, the injection volume was 100 uL, and the detector and column temperature were 50 °C.

#### 2.3.7. Hot Paste Viscosity Testing

The hot paste viscosities of the native starch and grafted starches were measured with a Rotational Viscometer (Shanghai Pingxuan Scientific Instrument Co. Ltd., Shanghai, China). Starch slurry with a mass percentage of 6% was prepared by dispersing starch (4.2 g dry basis) in 65.8 g distilled water and then heating in a 95 °C constant-temperature water bath. The rotational viscometer was connected to the thermostat and gradually increased to 95 °C, then maintained for 15 min. The viscosity value was recorded when the viscometer became stable.

#### 2.3.8. Scanning Electron Microscope (SEM) Analysis

The morphology of the native starch and grafted starches was determined with a scanning electron microscope (QUANTA 200, FEI, Amsterdam, the Netherlands), operating at an acceleration voltage of 20 kV. Starch granules were mounted on circular aluminum stubs with double-sided adhesive tape and coated with gold before testing.

#### 2.3.9. X-ray Diffraction (XRD) Analysis

The native starch and grafted starch samples were further dried in a vacuum oven at 50 °C for 48 h to remove the remaining moisture. The crystallinity index of the samples was measured by an X-ray diffractometer (XD-2, Beijing’s General Instrument Co., Ltd., Beijing, China) with Cu target at 36 Kv and 20 mA. Samples were tested in the angular range of 2θ = 5–40° with a scanning rate of 4°/min.

#### 2.3.10. Thermogravimetric (TGA) Analysis

TGA measurements of native starch and grafted starch samples were made with a 209 F3 TGA instrument (NETZSCH Co., Bavaria, Germany). About 5 mg of dried sample powders were placed in a platinum crucible and heated from 25 to 800 °C at the rate of 10 °C/min. Dynamic carrier nitrogen gas flowed at a rate of 30 mL/min. Thermogravimetric (TG) and derivative thermogravimetric (DTG) data were obtained for each sample.

#### 2.3.11. Differential Scanning Calorimeter (DSC) 

The gelatinization temperature of NS, DS and ATDS were studied by using a Differential Scanning Calorimeter (NETZSCH D204, Selb, Germany), as described by Sandhu and Singh [23]. Starch (3.5 mg, dry weight) was loaded into an aluminum pan and distilled water was added with a microsyringe to achieve a starch‒water suspension containing 70 g starch and 100 g water. Samples were hermetically sealed and allowed to stand for 1 h at room temperature before heating in a DSC. The DSC analyzer was calibrated using indium and an empty aluminum pan was used as a reference. Sample pans were heated at a rate of 5 °C/min from 30 to 200 °C.

#### 2.3.12. Statistical Analysis 

The data in the current study were statistically evaluated using the statistical software package Minitab Version 15 (Minitab Inc., State College, PA, USA) and reported as the mean value ± standard deviation of the replicates. A single-factor analysis of variance was conducted to identify significant differences among mean values according to least significant difference criteria with a 95% confidence level (*p* < 0.05).

## 3. Results and Discussion

### 3.1. Graft Reaction Confirmation

Figure 1 depicts the graft reaction between hydrophobic monomers and native corn starch. As shown, the successful reaction adds a new functional group (C=O) to the starch molecule. FTIR analysis of native starch and grafted starches was performed to verify that the graft reaction occurred and to investigate the resulting chemical changes. The results of this analysis are shown in Figure 2.

The basic compositional unit of native corn starch is d-anhydroglucose, with C2 and C3-linked secondary hydroxyls as the main characteristic functional groups, a C6-linked primary hydroxyl, and a d-pyranose ring structure [24]. The positions of the absorption peaks in the infrared spectrum for these main structures are shown in Figure 2: the characteristic peak centered at 3310 cm^−1^ corresponds to O‒H stretching and vibration of the hydrogen bond association, 2930 cm^−1^ corresponds to C‒H asymmetrical stretching and vibration, 1635 cm^−1^ arises from the water that is tightly bound to the starch, 1152 cm^−1^ is from C‒O‒C asymmetrical stretching and vibration, 1080 cm^−1^ corresponds to d-glucopyranose and hydroxyl-linked C‒O stretching and vibration, and 925 cm^−1^ is due to glucosidic bond vibration [25]. The infrared spectra for the grafted starches include all of the above characteristic absorption peaks, but also a C=O absorption peak at 1720 cm^−1^ [26,27]. Following the reaction of the starch with hydrophobic monomers, the unreacted hydrophobic monomers and homopolymer were removed by the acetone wash. It could be confirmed that the C=O came from grafted starches according to the positions that appeared, which verified that the graft reaction had occurred between the native starch and grafting monomer.

To further validate that the grafting reaction had occurred, the samples were subjected to X-ray photoelectron spectroscopy (XPS) to establish the binding modes of C in the samples; the results are shown in Figure 3 and Table 1. Compared with native starch, the relative concentration of C-containing groups with a binding energy between 284.9 and 289.0 eV was significantly different in the grafted starch samples. Comparing the XPS spectrum of native starch (Figure 3a) to the spectra of MAH-*g*-starch (Figure 3b), LA-*g*-starch (Figure 3c), and MA-*g*-starch (Figure 3d) shows that the intensity of the peak at 284.9 eV was decreased, which was attributed to the presence of more C‒C/C‒H groups. At the same time, the intensity of the peak at 286.1 eV was decreased, which represented the presence of C‒O groups. From the results in Figure 3, we see that the amount of C‒O in the grafted starches was higher than that in the native starch, which was attributed to the high incidence of grafted molecular chains in the grafted starch material. The intensity of the 287.6 eV peak was also changed, which reflected the presence of C‒C=O groups. In addition, the intensity of the 289.0 eV peak for the modified starch was changed, which was attributed to the abundance of O=C‒O groups. The chemical environmental changes indicated that the corn starch had reacted with the hydrophobic monomer through in situ solid phase polymerization, in which the C‒O(H) bond was broken and carbonyl groups (C=O) were generated. The results of the relative content of each chemical structure indicated that there was a difference in the content of C=O in the three kinds of grafted starches. The content of C=O in native starch was 15.32%; after the treatment, the C=O content increased to 16.86% in MAH-*g*-starch, 17.69% for LA-*g*-starch, and 15.69% for MA-*g*-starch. This difference in the three treated starches may be due to different reaction efficiencies, resulting in differences in the grafting ratio. To verify these differences, the grafting ratio was tested for the three kinds of graft starches.

### 3.2. The Grafting Ratio for Different Hydrophobic Monomers

The reaction variable of the in situ solid polymerization reaction between native starch and hydrophobic monomer was directly related to the variety of hydrophobic monomer, and inevitably leads to the difference in the grafting ratio. At the same time, the graft copolymerization of starch was carried out by the aqueous phase polymerization and organic solvent polymerization methods, and the grafting ratio has been tested; the results are shown in Table 1.

As shown in Table 1, compared with in situ solid phase polymerization, the degree of substitution and grafting ratio of aqueous phase polymerization were relatively small. This was due to the presence of hydrolytic side reactions in aqueous phase, resulting in a smaller grafting ratio. The degree of substitution and grafting ratio of grafted starches prepared by organic solvent polymerization was also smaller than those of in situ solid phase polymerization. It was indicated that starch and graft monomers had the highest reaction efficiency through in situ solid phase polymerization. The reaction efficiency of different graft monomers was also different. Moreover, the change trend of degree of substitution and grafting ratio of the grafted starches prepared by different grafting methods were the same.

With the in situ solid phase polymerization, the grafting ratio of MAH-*g*-starch was 6.50% ± 0.42%, that of LA-*g*-starch was 12.45% ± 0.37%, and MA-*g*-starch was 0.57% ± 0.05%. The results show that the efficiency of in situ solid phase polymerization for lactic acid and starch was the highest under these conditions. The XPS analysis showed that the C=O content of LA-*g*-starch was the highest and that of MA-*g*-starch was the smallest. The in situ solid phase polymerization of lactic acid and maleic anhydride with starch occurred as esterification, while methyl acrylate occurred as a radical polymerization. The reaction conditions required for radical polymerization are more stringent, resulting in the lowest grafting ratio for MA-*g*-starch. Lactic acid is a liquid that can penetrate into the starch well before the reaction, increasing the chance of contact with hydroxyl groups on the starch. In contrast, maleic anhydride is solid, and does not penetrate into the starch to react with hydroxyl until the temperature exceeds its melting point (52.8 °C). Additionally, maleic anhydride needs to open the acid anhydride to react with hydroxyl groups. These two reasons cause the grafting ratio of the LA-*g*-starch to be significantly greater than that of the MAH-*g*-starch.

The effects of reaction temperature and reaction time on the grafting ratio of three kinds of grafted starches were discussed. The results are shown in Figure 4. With the increase in reaction temperature, the grafting ratio of three kinds of grafted starches all increased gradually, and reached the maximum at 80 °C. However, when the reaction temperature was higher than 80 °C, the grafting ratio began to decrease. The temperature was too high; the molten monomers evaporate into the reaction vessel in the form of vapor. As the distance between monomers’ gasification and starch increased, the collision probability decreased, and the reaction efficiency also decreased. In addition, when the temperature was too high, the possibility of the homogenization of monomers increased, resulting in a decrease in the amount of monomers and a decrease in the grafting ratio. The reaction time was too short and the grafting reaction between monomers and starch was incomplete. At this time, the grafting ratio of three kinds of grafted starches was lower. When the reaction time was extended from 0.5 to 2.0 h, the grafting ratio gradually increased. However, when the reaction time exceeded 2.0 h, the grafting ratio had basically stabilized. This was due to the fact that the graft copolymerization of starch with monomers tends to be saturated.

### 3.3. Relative Hydrophobicity Change of Grafted Starches

The molecular chain of the native starch contains many hydrophilic hydroxyl groups, and the grafting of hydrophobic groups onto the native starch would improve the relative hydrophobicity of the starch. The contact angle (CA) of water on a surface is the angle formed by a tangent line from the water droplet to the solid surface, and is an indication of the relative hydrophobicity of the sample surface. Generally speaking, the larger the CA, the higher the water resistance of the material [28]. The surface contact angles of the native starch and the modified starches were measured with a contact angle tester to determine the relative hydrophobicity, and the results are shown in Figure 5. As shown by the data presented in Figure 5, the initial contact angle of the native starch was only 37° and full absorption of water droplets required only 1.010 s. After the modification of in situ solid phase polymerization, the initial contact angle was increased and the absorption time of water droplets was prolonged for all grafted starches. The results suggested better relative hydrophobicity of the grafted starches compared to native starch, due to the replacement of the hydrophilic hydroxyl groups on the bamboo fiber with hydrophobic carbonyl groups. The contact angle and the time of full absorption varied for the three kinds of grafted starches. The LA-*g*-starch had the largest contact angle and the longest absorption time and the MA-*g*-starch had the smallest contact angle and the shortest absorption time, indicating that the relative hydrophobicity of LA-*g*-starch was the best and that of MA-*g*-starch was the worst. The relative hydrophobicity of the grafted starches was directly related to the number of hydrophobic groups grafted into the starch, and more hydrophobic groups resulted in better relative hydrophobicity. Thus, the measured contact angles were consistent with the grafting ratios.

To further confirm that graft modification can improve the relative hydrophobicity of native starch, the water absorption of the materials was determined based on their relative weight change after exposure to water. The results are shown in Table 2. After incubation of the samples in a wet environment for 144 h, the weight gain rate of the samples became stable, as saturation was reached. The water absorption of the grafted starches was lower than that of native starch for the 144 h of measurement. Again, the results indicated that reaction between the hydrophobic monomer and native starch enhanced the relative hydrophobicity of the starch. Comparison of the water absorption for the three kinds of grafted starches revealed that LA-*g*-starch was the lowest and MA-*g*-starch was the highest. Thus, the LA-*g*-starch had the best relative hydrophobicity, and the MA-*g*-starch had the worst. The results were in agreement with the contact angle test results.

### 3.4. Molecular Weight Change of Grafted Starches

As shown in Figure 1, the graft copolymerization can increase the molecular weight of native starch. To verify the feasibility of the in situ solid polymerization reaction, native starch, MAH-*g*-starch, LA-*g*-starch and MA-*g*-starch were characterized by gel permeation chromatography (GPC) and the results are shown in Table 2. 

It can be seen from Table 2, the number-average molecular weight (*M*_n_) of native starch was 7.869 × 10^4^ D, the weight average molecular weight (*M*_w_) was 4.325 × 10^5^ D, and the distribution index (DI) was 5.496. Compared with native starch, the *M*_n_ and *M*_w_ of grafted starches increased and the DI increased also. It was also proved that the grafting monomers were successfully grafted onto the starch molecular chain. However, the *M*_n_, *M*_w_ and DI of grafted starches obtained by different grafting monomers was different. The *M*_n_ and *M*_w_ of from large to small was LA-*g*-starch, MAH-*g*-starch, MA-*g*-starch. The molecular weight distribution index also showed the same trend. This phenomenon was closely related to the grafting ratio of grafted starches. In Table 1, the grafting ratio of LA-*g*-starch was the largest, the polylactic acid molecular chain grafted on the starch molecular chain was the most. But the grafting ratio of MA-*g*-starch was only 0.57%, which leads to the minimum molecular weight and distribution index.

For polymers, viscosity was proportional to molecular weight. In order to further verify the effect of the in situ solid phase polymerization on the molecular weight of modified starches, the pasting viscosity of the native starch and grafted starches were tested. The pasting viscosity of the native starch was 16,500 mPa·s. Compared with the native starch, the pasting viscosity of all grafted starches were increased. The pasting viscosity of MAH-*g*-starch, LA-*g*-starch and MA-*g*-starch were 23,600 mPa·s, 27,100 mPa·s and 17,200 mPa·s, respectively. It was proved that the molecular weight of starches increased after graft copolymerization. The pasting viscosity of LA-*g*-starch was the largest and MA-*g*-starch was the smallest, which indicated that the average molecular weight of LA-*g*-starch was the largest and that of MA-*g*-starch was the smallest. This also validates the results of the GPC test.

### 3.5. Morphology Change of Grafted Starches

Scanning electron microscopy, in principle, was to use a very fine focused high-energy electron beam to scan the sample and stimulate a variety of physical information, by accepting, amplifying and displaying this information, the surface morphology of the test specimen was observed. The various samples of test materials were subjected to SEM analysis to determine the extent of any changes in surface morphology of the material as a result of the grafting reaction.

Figure 6 shows the SEM micrographs of the grafted starch, where it can be seen that the native starch granules were solid circles with smooth surface and edges. Comparison with the native starch, the in situ solid phase polymerization did not destroy the granule structure of the starch. This may be because solid phase reaction process was conducted under anhydrous conditions, and there was no starch gelatinization. There were morphological changes in the grafted starches, as the surface became rough and the granule surface was destroyed. This was because the graft polymer was grown on the starch granules, and the granules were bonded together. However, there was no obvious change in granule size, indicating that in situ solid phase polymerization mainly occurred on the surface of the starch granule. The SEM micrographs showed that the surface roughness and bond degree of MAH-*g*-starch and LA-*g*-starch were more obvious, especially for the LA-*g*-starch. The results indicate that the surface roughness of the grafted starch was positively correlated with the grafting ratio.

### 3.6. Crystalline Structure Change of Grafted Starches

The FTIR, XPS, and SEM analysis of grafted starches all verified that a grafting reaction occurred as a result of the in situ solid phase polymerization method. Therefore, it was reasonable to assume that the crystalline structure of the reactants had also changed. We next analyzed the crystalline structure of the native starch and the grafted starches to further verify this conclusion. The XRD method was used to determine if there was any change in the crystal structure of the native starch as a result of the grafting reaction, and the results are shown in Figure 7.

The XRD diffraction peaks for native starch were typical of an A-type crystalline structure, with 2θ values of 15°, 17°, 18°, and 23° [29]. After the grafting reaction, the crystallization type of grafted starches did not change, suggesting that the reaction mainly occurred in the amorphous areas of the starch. The intensity of the XRD diffraction peaks in the hydrophobically modified starch was clearly weaker compared to those of the native starch. According to the report [30], it can be calculated that the degree of crystallinity of native starch was 27.93%, and after modification, the crystallinity of the grafted starches was reduced to varying degrees. This suggested that in situ solid phase polymerization could destroy the crystalline structure of starch to some extent. In other words, the hydrogen bonds between molecules were weakened, thus the thermoplasticity of grafted starches increased [31]. This phenomenon was due to infiltration of the hydrophobic monomers into the crystalline area to destroy the hydrogen bonding between the molecules in this region. At the same time, the hydroxyl groups on the starch chain reacted with the hydrophobic monomers, and the molecular chains gradually grew and crosslinked, which further destroyed the crystallinity of the starch. The decrease in the crystallinity of the grafted starches varied, as the MAH-*g*-starch decreased to 23.47%, LA-*g*-starch decreased to 22.29%, and MA-*g*-starch decreased to 24.69%. The grafting ratio of LA-*g*-starch was the highest and more hydroxyl groups on the chain of starch reacted, with the most serious destruction of hydrogen bonds. The grafting ratio of MA-*g*-starch was the lowest and the destruction of its crystalline structure was the smallest. More graft polymer molecular chains attached to the starch molecular chain resulted in a more loose structure.

### 3.7. Thermal Stability Changes of Grafted Starches

The thermal behavior of the native starch and grafted starch were analyzed by TGA, and the results are shown in Figure 8. The thermogravimetric curves in Figure 8 show three stages at 50~120 °C, 200~350 °C, and 400~600 °C. The first stage (50~120 °C) represented the evaporation of water from the starch. In the second stage (200~350 °C), native starch and grafted starches thermally decomposed, and the fastest rate of decomposition was observed in this stage. However, in the presence of the different hydrophobic monomers grafted on the starch, the rate of heat loss was also different. In the third stage (above 400 °C), the remaining material was decomposed to carbon by pyrolysis. The heat loss rate of the grafted starches was greater than that of native starch, and there was less decomposed residue. Comparing the residual weight, it appeared that the grafted starch materials had an increased weight loss rate (residual reduction), but the initial temperature for thermal decomposition and heat loss was reduced. This result also suggested an increase in the plasticity of the starch as a result of the grafting reaction. The XRD results showed that modification destroyed the starch crystallization and the density degree of the starch molecules decreased, so the grafted starch decomposed easily when heated. Of the grafted starch materials, the initial temperature of LA-*g*-starch was the lowest and the decomposition residual rate was the highest, and the MA-*g*-starch showed the opposite trends. The LA-*g*-starch was most easily decomposed, indicating the best thermoplasticity in the melting process. The most serious damage was evident for the crystalline structure of the LA-*g*-starch, and the MA-*g*-starch crystallinity was the lowest.

The DTG results in Figure 8 show a heat loss rate-accelerated peak for several kinds of starch that represents the maximum decomposition rate temperature. The maximum decomposition rate temperature of the native starch was 319.6 °C, but the grafted starches showed obviously lower values. This was due to the destruction of the crystalline structure of the starch by the modification of the in situ solid phase polymerization, reducing the decomposition temperature of the grafted starches. The results agreed with the results of TG analysis, and the maximum decomposition rate temperature of the LA-*g*-starch was the lowest and that of MA-*g*-starch was the highest.

### 3.8. Melting Property Changes of Grafted Starches

DSC is a common method for researching melting properties [32]. The starch melting can also be used to characterize the degree of difficulty for starch plasticization. By comparing the DSC endothermic peak area, the change of melting enthalpy after grafting modification by different monomers. The degree of gelatinization of native starch, MAH-*g*-starch, LA-*g*-starch and MA-*g*-starch were tested by DSC, and the results are presented in Figure 9. The thermal transitions temperature of the samples were defined as *T*_o_ (onset), *T*_p_ (peak) and *T*_c_ (conclusion), and the enthalpy of melting was referred to as Δ*H*.

As can see in Figure 9, native starch showed *T*_o_ and *T*_p_ between 103.54 and 134.88 °C, *T*_p_ was 129.03 °C, and Δ*H* was 630.27 J/g. After in situ solid phase grafting modification of MAH, LA and MA, the *T*_o_, *T*_p_, *T*_c_ of grafted starches all shifted to lower temperature, and Δ*H* was decreased to varying degrees. It was indicated that the grafted starches was easier to gelatinization and lower in enthalpy of gelatinization. This was mainly attributable to the decrease of crystallinity and intermolecular loosening after graft modification. There were differences in gelatinization temperature and gelatinization enthalpy of three grafted starches. The *T*_o_, *T*_p_, *T*_c_, and Δ*H* of LA-*g*-starch were the lowest, and those of MA-*g*-starch were the highest. The reason for this difference was that the degree of crystallinity of the three kinds of grafted starches varies with different grafting ratios. At the same time, this indicates that LA-*g*-starch had the best thermoplasticity and MA-*g*- starch the worst.

## 4. Conclusions

This study demonstrated the successful preparation of three grafted starches using an in situ solid phase polymerization reaction. The grafting ratios of the MAH-*g*-starch, LA-*g*-starch, and MA-*g*-starch were 6.50%, 12.45%, and 0.57%, respectively. The grafted reaction resulted from replacement of the hydroxyl groups on d-anhydroglucose moieties of the starch with hydrophobic carbonyl groups form the graft monomer, which resulted in improved relative hydrophobicity of the starch. With the influence of the grafting ratio, LA-*g*-starch had the best relative hydrophobicity and MA-*g*-starch had the worst. The M_n_, M_w_, and DI of LA-*g*-starch was the largest and MA-*g*-starch was the smallest. The surface of the grafted starches was covered with graft polymer, and the surface roughness and the bond degree of the MAH-*g*-starch and LA-*g*-starch were more obvious. The starch crystalline structure was destroyed by the grafting reaction, the degree of crystallinity was decreased, and the hydrogen bonding between the starch molecules was weakened, resulting in improved thermoplasticity in the grafted starches. For the degree of crystallinity of grafted starches, MAH-*g*-starch decreased to 23.47%, LA-*g*-starch was 22.29%, and MA-*g*-starch was 24.69%. The degree of thermoplasticity improvement from large to small was LA-*g*-starch, MAH-*g*-starch, and MA-*g*-starch.

The grafted starches produced by in situ solid phase polymerization exhibited increased overall relative hydrophobicity with increased interface compatibility, which allows for expanded application of thermoplastic starch and starch/plastic composites. This work comparing the influence of three grafting monomers on the grafting ratio and hydrophobic modification of starch provides reference data for the preparation of blended composites of starch and different polymers.

## Figures and Tables

**Figure 1 polymers-11-00072-f001:**
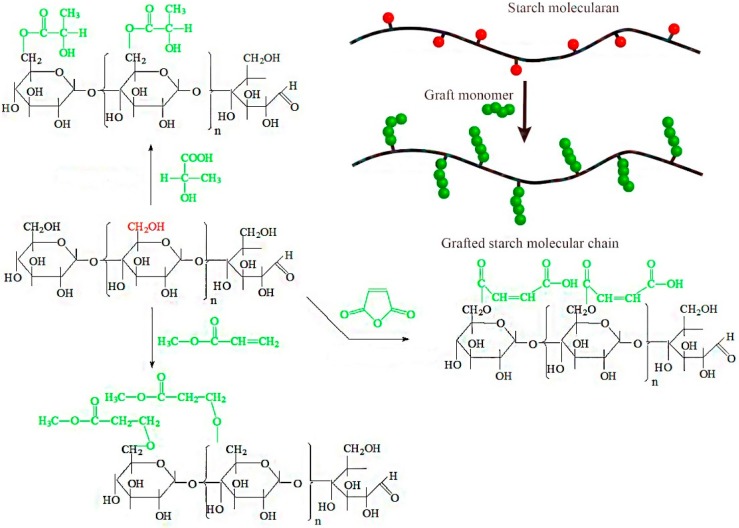
In situ solid phase polymerization of corn starch and hydrophobic groups.

**Figure 2 polymers-11-00072-f002:**
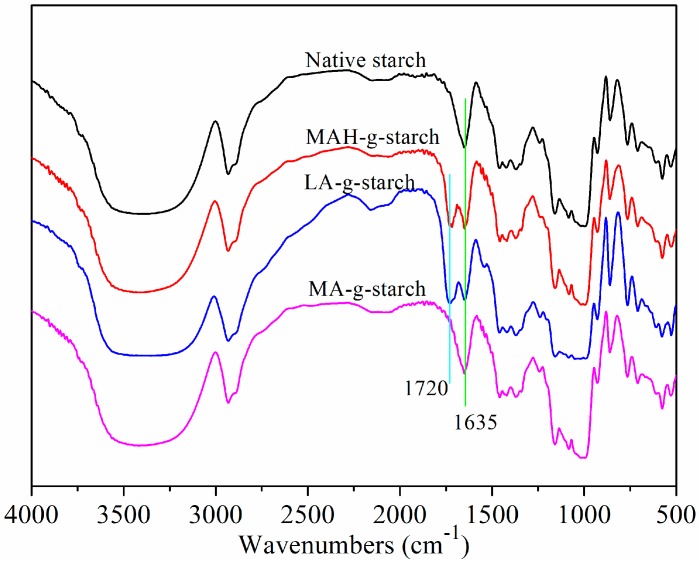
Infrared spectrum of native starch and grafted starches.

**Figure 3 polymers-11-00072-f003:**
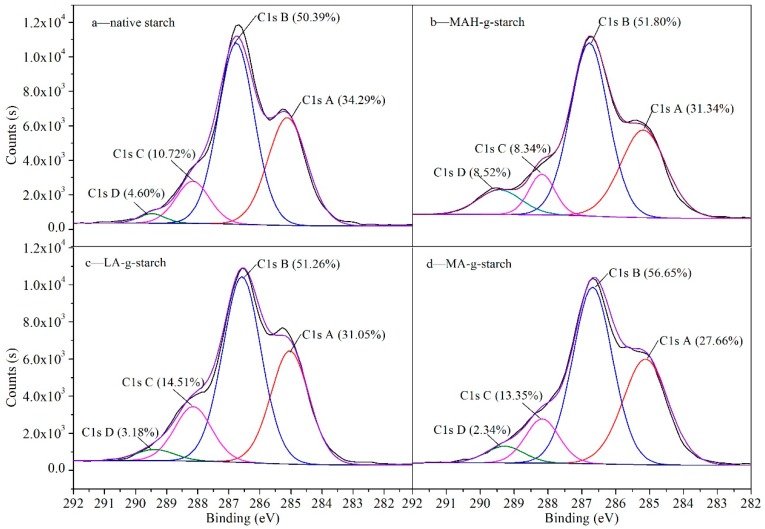
The binding modes and relative content of C in native starch and grafted starches.

**Figure 4 polymers-11-00072-f004:**
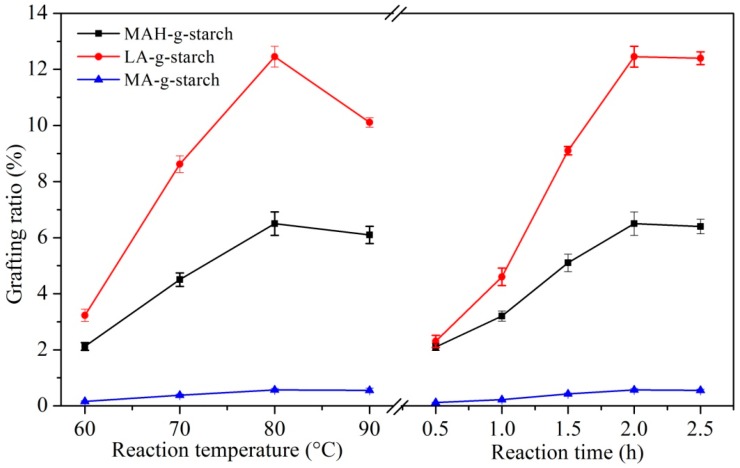
Effect of reaction temperature and reaction time on grafting rate of grafted starches.

**Figure 5 polymers-11-00072-f005:**
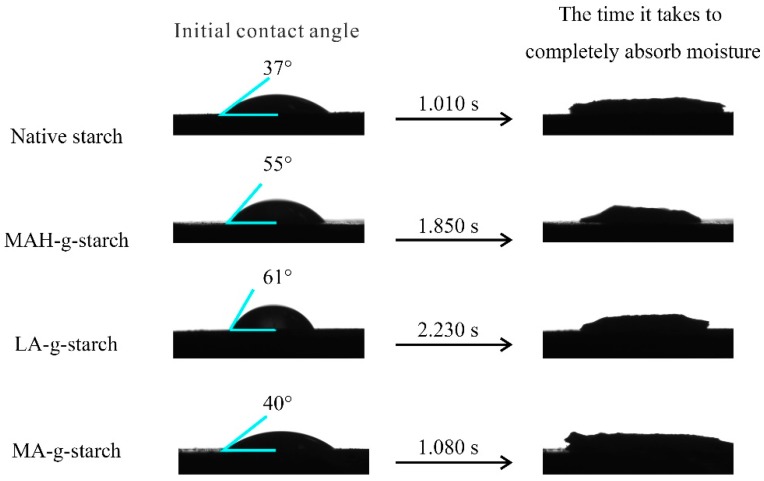
Initial surface contact angle for the native starch and grafted starches.

**Figure 6 polymers-11-00072-f006:**
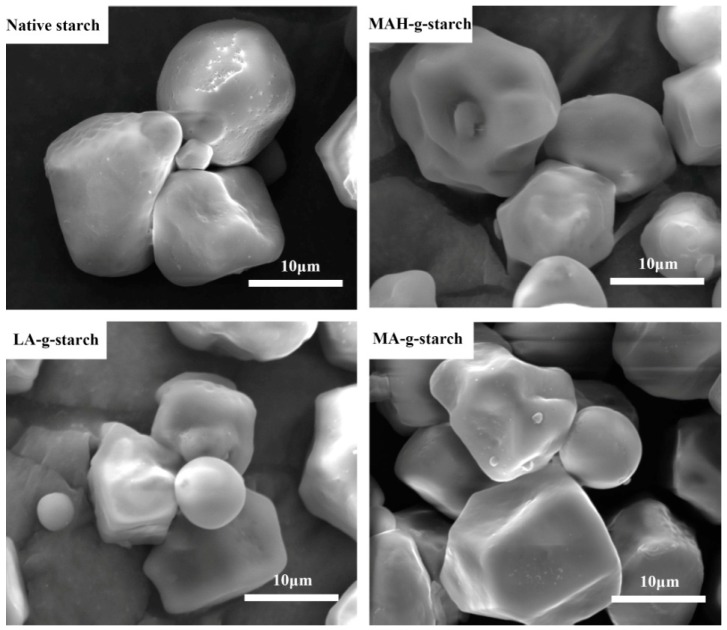
SEM images of native starch and grafted starches.

**Figure 7 polymers-11-00072-f007:**
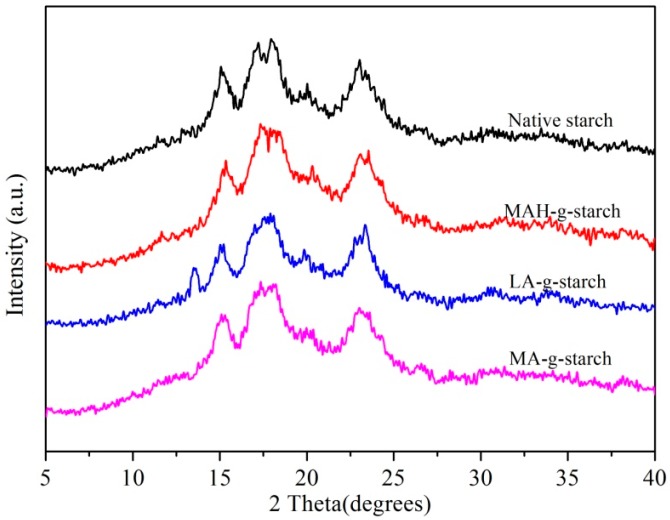
XRD diffraction patterns of native starch and grafted starch.

**Figure 8 polymers-11-00072-f008:**
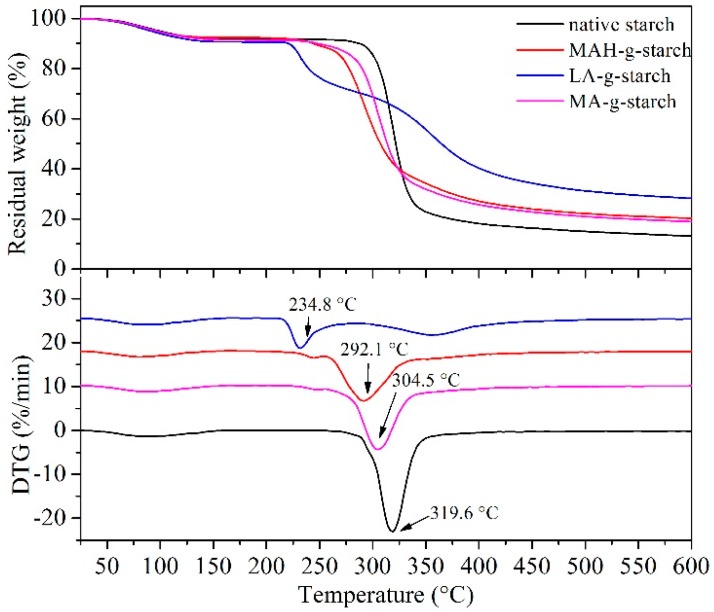
TGA and DTG curves of native starch and grafted starches.

**Figure 9 polymers-11-00072-f009:**
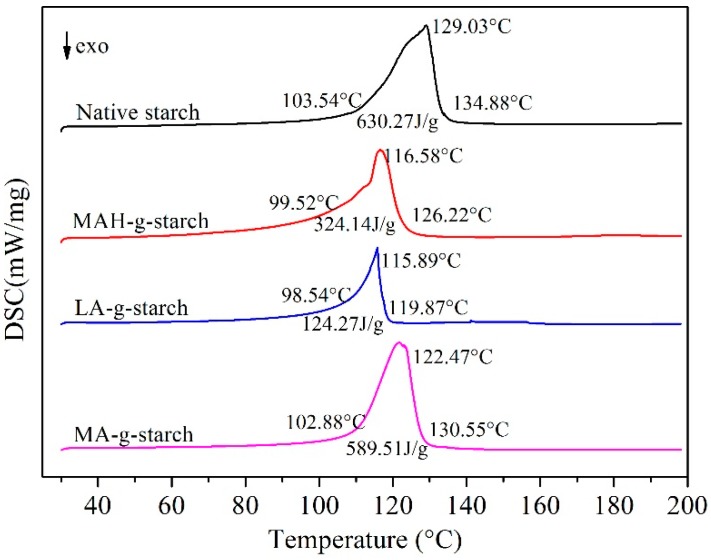
DSC curves of native starch and grafted starches.

**Table 1 polymers-11-00072-t001:** Grafting ratio of three kinds of grafted starches.

Grafted Starches	In Situ Solid Phase Polymerization		Aqueous Phase Polymerization		Organic Solvent Polymerization	
	*W* (%)	*GR* (%)	*W* (%)	*GR* (%)	*W* (%)	*GR* (%)
MAH-*g*-starch	3.79 ± 0.23	6.50 ± 0.42	1.27 ± 0.07	2.13 ± 0.11	3.58 ± 0.21	6.14 ± 0.30
LA-*g*-starch	6.47 ± 0.28	12.45 ± 0.37	3.06 ± 0.09	5.68 ± 0.16	5.91 ± 0.32	11.31 ± 0.51
MA-*g*-starch	0.33 ± 0.04	0.57 ± 0.05	0.12 ± 0.02	0.23 ± 0.03	0.32 ± 0.04	0.60 ± 0.06

**Table 2 polymers-11-00072-t002:** Water absorption change and molecular weight of native starch and grafted starches.

Starch Species	Water Absorption (%)						Molecular Weight		
	24 h	48 h	72 h	96 h	120 h	144 h	*M*_n_ (D)	*M*_w_ (D)	DI
Native starch	20.175	22.679	24.649	26.680	27.556	27.584	7.869 × 10^4^	4.325 × 10^5^	5.496
MAH-*g*-starch	16.487	19.060	20.932	22.973	23.992	24.389	8.422 × 10^4^	7.528 × 10^5^	8.938
LA-*g*-starch	12.981	15.288	17.112	18.856	19.369	19.554	1.726 × 10^5^	2.141 × 10^6^	12.405
MA-*g*-starch	17.423	20.320	22.823	25.069	26.486	26.717	8.014 × 10^4^	4.812 × 10^5^	6.004

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
