# Peer review of "Preparation and Characterization of Hydrophobically Grafted Starches by In Situ Solid Phase Polymerization"

_polymers, 2019, doi:10.3390/polym11010072_

Round 1

Reviewer 1 Report

This manuscript is about preparation of hydrophobic groups (MAH-g-starch; LA-g-starch;MA-g-starch) grafted starches by in-situ solid phase, aqueous phase polymerization method and organic solvent polymerization method. This work needs to be improved to be published in Polymers, the improvements are:

- Review the grafting ratios of MAH-g-starch (6.5%), LA-g-starch (12.45%) and MA-g-starch (0.57%), are very low values, the authors should compare with existing values in the literature.

- In preparation of hydrophobically modified starch, the reaction temperature and reaction time on the grafting degree should be investigated in the 3 methods.

- In hydrophobic property of grafted starches, this section has to be rewritten, because to be hydrophobic, the contact angle must be greater than 90º 

- The SEM micrographs should be improved, the authors should obtain images with greater magnification to see the surface of starch and grafted starches

- How was the crystallinity calculated?

Reviewer 2 Report

In this manuscript the authors reacted starch with three hydrophobic groups.  Whereas the study is of interest, improvements are needed before this manuscript can be published.

 The title sounds awkward.  It may be better to say, "Preparation and characterization of hydrophobically grafted starch by in-situ solid phase polymerization."

In p.3 the authors reviewed some hydrophobically modified starch. However, they ignored starch acetate (acetylated starch).  This should be included in their text. 

From the description, it seems to me that only limited number of samples were synthesized for this paper.  Since the authors places a great emphasis on their solid phase polymerization, it is surprising that they did not make more samples and study the dependence of the process variables on the polymer characteristics.  For example, what is the effects of temperature, reaction time, catalyst level?  What is the yield of the reaction?  These are important synthetic considerations that should not be omitted.  If the authors have not done these studies, they should. This is also useful information for other people to repeat this work. 

I am surprised that the authors did not use solution NMR to characterize the products. NMR (especially 13C) provides a lot more informative than IR and XPS. When properly done, it is also quantitative and can give the degree of substitution with much better accuracy.  With good analysis, sometimes you can get the average chain lengths of the graft polymers. In fact, in polymer (and organic) syntheses NMR analysis is considered routine. 

The caption for Figure 3 should indicate XPS as the nature of the data.

In p. 7 (Table 1), the authors claimed that solid phase polymerization gave the highest reaction efficiency. Yet, the data in Table 1 suggest that organic solvent polymerization gave comparable values.  They should also indicate the error ranges of the values reported.  (I presume these reactions were done more than once to ensure reproducibility, so the average and the standard deviation should be reported for each entry in the table.)

In pp. 11-12, the topic titles for 3.7 and 3.8 do not sound right. What is thermal "performance" analysis?  Also "thermoplasticity" means more than melting.  These should be reworded.

Round 2

Reviewer 1 Report

The revised manuscript is ok for me